# Adolescents' reports of chaos within the family home environment: Investigating associations with lifestyle behaviours and obesity

Andraea Van Hulst[1]*, Sujani Jayanetti[1], Ana Maria Sanson-Rosas[1], Marie-Josée Harbec[2], Lisa Kakinami[3,4], Tracie A. Barnett[5,6], Mélanie Henderson[6,7,8]

**1** Ingram School of Nursing, McGill University, Montréal, QC, Canada, **2** École de Psychoéducation, Université de Montréal, QC, Canada, **3** Department of Mathematics and Statistics, Concordia University, Montréal, QC, Canada, **4** PERFORM Centre, Concordia University, Montreal, QC, Canada, **5** Department of Family Medicine, McGill University, Montréal, QC, Canada, **6** Centre de Recherche du CHU Sainte-Justine, Montréal, QC, Canada, **7** Department of Pediatrics, Université de Montreal, Montréal, QC, Canada, **8** School of Public Health, Department of Social and Preventive Medicine, Université de Montréal, Montreal, QC, Canada

* andraea.vanhulst@mcgill.ca

**Data Availability Statement:** For ethical reasons, data from study participants cannot be shared openly as they include potential identifying

## Abstract

### Objective

Disorganised and chaotic home environments may hinder the adoption of healthy lifestyle behaviours and contribute to excessive weight gain among adolescents. We examined whether self-reported level of chaos within the family home environment is associated with lifestyle behaviours and obesity in adolescent girls and boys.

### Methods

Cross-sectional data from the 3rd wave of the Québec Adipose and Lifestyle Investigation in Youth (QUALITY) study were analyzed. The sample consisted of n = 377 White adolescents with a history of parental obesity. Home environment chaos was measured using the Confusion, Hubbub, and Order Scale (CHAOS) analyzed both continuously and dichotomized as high vs. low chaos. Body Mass Index z-scores (zBMI) were computed using WHO standards from measured weight and height. Physical activity (7-day accelerometry), vegetable and fruit intake (three 24-hour diet recalls), and leisure screen time and sleep duration (questionnaire) were assessed. Sex-specific linear regression models were used to estimate associations between level of family home environment chaos, lifestyle behaviours and zBMI.

### Results

The overall level of chaos was low in our study sample, with higher reported levels among girls compared to boys. Among girls, high (vs low) chaos was associated with shorter sleep duration (hours/day) (B = - 0.44, 95% CI: -0.75, -0.14). No associations were observed for other lifestyle behaviours or for zBMI.

participant information. Moreover, participants have not provided consent for data to be deposited in a public repository. This statement was validated with the Research Ethics Board that provided initial approval for the QUALITY Cohort study, that is the Research Ethics Board of the CHU Sainte-Justine Hospital presided by Me Geneviève Cardinal (genevieve.cardinal.hsj@ssss.gouv.qc.ca).

**Funding:** The QUALITY study (primary data collection) was funded by grants from CIHR (https://cihr-irsc.gc.ca/e/193.html) (#OHF-69442, #NMD-94067, #MOP-97853, #MOP-119512), HSFC (https://www.heartandstroke.ca) (#PG040291), and FRQS (https://frq.gouv.qc.ca). AVH and LK hold a Fonds de la recherche en santé du Québec (FRQS) Junior 1 award and MH holds a Junior 2 award from the same organization. The secondary analysis presented herein did not receive any funding. The funders had no role in study design, data collection and analysis, decision to publish, or preparation of the manuscript.

**Competing interests:** The authors have declared that no competing interests exist.

## Conclusion

In this sample of adolescents with a parental history of obesity, higher household chaos was not associated with obesity or lifestyle behaviours, except for sleep duration among girls. Replication of findings in more diverse samples is indicated.

## Introduction

Overweight and obesity is a global public health concern with high prevalence among children and adolescents [1]. In Canada, 34.4% of adolescents aged 12 to 17 years are affected by this condition [2]. Excess body weight in childhood and adolescence tends to be maintained into adulthood and is associated with increased morbidity and mortality [3–5]. Overweight/obesity is a complex condition in which many factors across multiple levels of influence are implicated. The Ecological Systems Theory recognises that characteristics of the family home environment can protect against or promote excessive weight gain in children and adolescents [6–8]. Specifically, chaos within the family home environment is gaining interest in relation to child socio-emotional, behavioural, and health outcomes [9–13], including childhood obesity [8, 9].

Household chaos is a complex construct of the family home environment encompassing two main dimensions, namely instability and turbulence (i.e., recurrent changes in residential location or family composition, lack of family routines), as well as disorganization (i.e., high levels of background noise, crowding, clutter, and a lack of structure) [14–17]. A growing body of research has documented the role of household chaos on obesity-related lifestyle behaviours among children. For example, among preschool-aged children and adolescents, household chaos has been found to mediate the association between lower socioeconomic status and lower sleep quality [18, 19]. Chaotic home environments have also been associated with higher screen time among pre-school aged children [20] and among rural school-aged children [21]. Another study reported that during the COVID pandemic, higher household chaos was associated with lower physical activity and sleep duration, and with higher screen time in preschoolers [22]. Similarly, higher chaos in the family home environment has been associated with unhealthy eating behaviours in young children, such as increased fat intake [23] and lower frequency of family meals, as well as with negative perceptions surrounding meal preparation among parents [24].

Despite the compelling evidence exposing a relation between chaotic family home environments and less healthy lifestyle behaviours, household chaos has been inconsistently associated with childhood overweight/obesity [17, 25]. Although some studies report direct effects of household chaos on child weight status [26], others report indirect effects [27] or no effects [28, 29]. One study among adolescents entering a weight management program found higher chaos in the home environment to be associated with higher baseline body mass index and lower short term success in weight loss [30]. Moreover, sex differences in the association between chaotic home environments and child weight status have been reported in one study of toddlers where an association was found only among boys but not among girls [25].

The vast majority of studies on household chaos, lifestyle behaviours and weight status in children have been conducted among preschool-aged children and there is a paucity of studies analyzing this association in adolescents [8, 9]. Moreover, very few studies have explored sex or gender differences in these associations. To contribute to fill these knowledge gaps, we examined whether adolescent reported household chaos is associated with obesity and with

related lifestyle behaviors, namely physical activity, sleep duration, screen time and vegetable and fruit intake, and explored sex differences in these associations. We hypothesised that higher household chaos would be associated with poorer lifestyle behaviours and higher obesity. We did not have any predetermined hypothesis on sex differences given the exploratory nature of these associations.

## Materials and methods

### Study design and participants

Cross-sectional data of the 3rd wave of data collection from the Quebec Adipose and Lifestyle Investigation in Youth (QUALITY) cohort study were analysed. QUALITY is an ongoing longitudinal investigation on obesity and cardiovascular risk factors among children aged 8–10 years at baseline (N = 630). Participants were recruited from elementary schools located in 3 major urban centers in Québec, Canada. Eligibility required participants to be White and both parents had to be available to participate at baseline with at least one parent having obesity (i.e., BMI $\geq$30 kg/m$^2$ and/or waist circumference >102 cm in men and >88 cm in women). Detailed information about the QUALITY cohort can be found elsewhere [31]. Two follow-up assessments were conducted when the participants were aged 10–12 years and 15–17 years. The current analysis is restricted to the 377 adolescents who completed the 3rd wave of data collection (60% retention from the baseline evaluation). Written consent and assent were provided by the parents and adolescents, respectively. Ethics approval was obtained for the QUALITY study from the Ethics Review Boards of the CHU Sainte Justine and the Québec Heart and Lung Institute, and from the McGill University Faculty of Medicine and Health Sciences Institutional Review Board for the current secondary data analysis.

### Data collection procedures

Data collection for the second follow-up visit was completed between 2012 to 2016 at the CHU Sainte-Justine Clinical Research Unit in Montréal and at the Quebec Heart and Lung Institute in Quebec City. Standardized procedures for anthropometric and lifestyle behaviour measurements were used, and questionnaires were completed by participants using a computerized form.

Household chaos was self-reported by adolescents using the Confusion, Hubbub, and Order Scale (CHAOS) [32], the most frequently used tool for the assessment of household chaos [9]. The original CHAOS questionnaire has been found to be an accurate and cost-effective tool to measure parent-reported environmental chaos in the household with satisfactory internal consistency (Cronbach's alpha = 0.79) and test-retest reliability [32]. In the current study, adolescents were asked to score the 15 CHAOS items using 5-point Likert scales ranging from definitely false to definitely true. Positive items were reverse-coded, and responses were then recoded as 0 (for definitely false, false, not really true or false) or 1 (for true, definitely true) to better reflect the original tool in which only two answer options are used (true or false). Consistent with the original tool, this resulted in a total score ranging from 0 to 15 where higher scores correspond to more chaotic households. In the current sample, Cronbach's alpha was found to be acceptable albeit lower than that reported elsewhere [32] (S1 Table). We examined CHAOS both as a continuous variable and as a dichotomized variable at the 75th percentile corresponding to higher versus lower CHAOS.

Weight was measured with an electronic scale with participants wearing light clothing [31]. Participants were weighed twice to the nearest 0.1 kg and if there was a difference of 0.2 kg or more, participants were weighed for a third time, and the average of the two closest values was used. Participants' height was measured with a stadiometer, without shoes [31]. The

measurement was taken at the time of maximum inspiration and recorded to the nearest 0.1 cm. Height was measured twice and if a difference of 0.2 cm or more was found a third measurement was taken, and the average of the two closest values was used. Body mass index (BMI) was calculated as weight (kg) divided by height squared ($m^2$) and was transformed to age- and sex-specific BMI Z-scores (zBMI) using WHO reference values and standard cut-offs for weight status categories (zBMI >1 for overweight and >2 for obesity) [33, 34].

Moderate-to-vigorous physical activity (MVPA) was assessed with an Actigraph monitor (Triaxial; GT3X, Actigraph LLC, Pensacola, FL, USA), a valid and reliable tool to measure physical activity [35]. Adolescents were provided with an accelerometer and instructed to wear the activity monitor for 7 days following the research visit. The data were included only for those participants who wore the activity monitor for at least 4 days and at least 10 hours per day [36] and underwent standardized quality control procedures and data reduction methods [37]. MVPA was calculated by adding the total number of minutes of daily moderate (defined as 2296 to 4011 counts per minute) and vigorous (defined as $\geq$4012 counts per minute) physical activity per day averaged over the total valid days of wear [38]. Physical activity was also dichotomized based on whether or not the participant met current recommendations of engaging in at least 60 minutes of MVPA per day [39].

Sleep duration was self-reported by adolescents using a questionnaire to document the typical school-day bedtime and wake-up time, and non-school day bedtime and wake-up time. Daily mean hours of sleep duration was then computed, as well as a dichotomous sleep duration variable based on whether or not participants met recommendations of at least 8 hours of sleep per night [39].

Screen time was self-reported by adolescents in a questionnaire assessing daily hours of television viewing, leisure computer, and video game use during a typical weekday and weekend day [40]. The weighted average daily hours for screen time was computed. Screen time was also dichotomized as meeting or not recommendations of no more than 2 hours of leisure screen time per day [39].

Dietary intake was collected by a trained dietitian and measured using mean values obtained from three 24-hour diet recalls on non-consecutive days including one weekend day [41]. Diet recall interviews were done by telephone, within a 6-week period following the research visit, with the adolescent and parent who prepared the meals [42]. The food ingested by adolescents were entered into CANDAT Nutrient Analysis software (Godin, London, Ontario) to determine participants' nutritional intake for the total food ingested or by food category. Daily average servings of vegetables and fruits intake was considered in this study as an indicator of overall diet quality [43]. The 2007 Canada Food Guide was used as a reference to compute the vegetable and fruit intake which was also dichotomized as meeting or not 5 or more servings of vegetables and fruits per day [44].

Covariates included sex and age obtained by adolescent self-reported questionnaire, as well as total household income adjusted for the number of people living in the household, parental education (1 or 2 parents with university degree vs both parents with less than a university degree) and family structure (single vs dual parent family) obtained from the parent-completed questionnaire. Covariates were identified a priori based on potential confounders for the associations of interest.

## Statistical analyses

Statistical analyses were performed using IBM SPSS Statistics for Windows, version 26 (IBM Corp., Armonk, N.Y., USA). Descriptive statistics including means (standard deviations), medians (interquartile ranges), and proportions were computed. Multicolinearity between all

variables was assessed. Multiple linear regression models were used to estimate associations between each dependent variable (zBMI, MVPA, sleep duration, screen time, and fruit and vegetable servings) and the primary independent variable (i.e., CHAOS continuous and dichotomized) in distinct models. Given the skewed distribution of MVPA, this variable was transformed using: 100 x Ln (MVPA) [45]. Beta coefficients for MVPA thus represent the % of change in MVPA for a 1-unit increase in CHAOS treated continuously, or for higher vs lower CHAOS when dichotomized. Models were adjusted for participants' age, parental education, household income, and family structure. Interactions terms between sex and CHAOS were tested for each dependent variable, and final results are presented for the full sample as well as stratified by sex. Beta coefficients with 95% confidence intervals are presented, and p-value of < 0.05 are considered statistically significant. In sensitivity analyses, we estimated multiple logistic regressions for associations between CHAOS and outcome variables, which were dichotomized according to whether recommendations for each lifestyle behaviour were met and for normal weight vs overweight/obesity.

## Results

Participant characteristics are summarized in Table 1. Overall, household chaos was low in this sample with a median CHAOS of 2. Girls (25.4%) were more likely than boys (16.7%) to report higher household chaos. In covariate adjusted linear regressions, no associations were observed for the full sample (Table 2). An interaction by sex was found for the association between CHAOS as a continuous score and sleep duration (interaction term p = 0.019). In sex

**Table 1. Characteristics of participant, QUALITY cohort study, visit 3 (n = 377).**

| Characteristics | Full sample | Boys | Girls |
|---|---|---|---|
| | Mean ± SD, Median (IQR), or % (n) | | |
| Age, years | 16.8 ± 1.0 | 16.8 ± 0.9 | 16.8 ± 1.0 |
| Sex | - | 54.1 (204) | 45.9 (173) |
| Weight status by category | - | - | - |
| Normal weight | 60.2 (227)* | 56.9 (116) | 64.1 (111) |
| Overweight | 22.8 (86) | 23.0 (47) | 22.5 (39) |
| Obese | 17.0 (64) | 20.1 (41) | 13.3 (23) |
| zBMI | 0.8 ± 1.3 | 0.8 ± 1.3 | 0.7 ± 1.2 |
| Moderate-to-vigorous physical activity (mins/day) | 24.2 (13.6, 37.4) | 29.9 (17.6, 42.5) | 17.7 (11.4, 30.3) |
| Leisure screen time (hours/day) | 4.5 ± 2.6 | 5.0 ± 2.6 | 3.9 ± 2.5 |
| Sleep duration (hours/day) | 8.8 ± 0.9 | 8.8 ± 0.9 | 8.9 ± 0.9 |
| Vegetable and fruit intake (servings/ day) | 4.7 ± 2.6 | 4.6 ± 2.7 | 4.8 ± 2.6 |
| CHAOS score | 2.0 (1.0, 3.0) | 1.0 (1.0, 3.0) | 2.0 (1.0, 4.0) |
| Low CHAOS (≤ 3) | 79.3 (299) | 83.3 (170) | 74.6 (129) |
| High CHAOS (> 3) | 20.7 (78) | 16.7 (34) | 25.4 (44) |
| Household income (Canadian $) | 57 040 ± 24 491 | 56 692 ± 23 709 | 57 448 ± 25 440 |
| Parental education | - | - | - |
| 1 or 2 parents with university degree | 55.7 (210) | 55.9 (114) | 55.5 (96) |
| Both parents with less than a university degree | 44.0 (166) | 43.6 (89) | 44.5 (77) |
| Family structure (lives with both parents) | 74.5 (281) | 75.0 (153) | 74.0 (128) |

Abbreviations: zBMI, body mass index z-scores based on the WHO reference norms; CHAOS, Confusion, Hubbub, and Order Scale; IQR, interquartile range.

*The normal weight status category includes 5 participants with an underweight BMI z-score.

For moderate-to-vigorous physical activity, data are missing for n = 53 participants, all other data are complete or missing for less than 2% of the sample

**Table 2. Associations (Beta, 95% CI) between household chaos, lifestyle behaviours and zBMI, in the full sample and stratified by sex, QUALITY cohort study, visit 3 (n = 377).**

| | zBMI | MVPA (% change in mins/day of MVPA) | Sleep duration (hours/ day) | Leisure screen time (hours/day) | Vegetable and Fruit intake (servings/ day) |
|---|---|---|---|---|---|
| **Full Sample (n = 377)** | | | | | |
| **MODEL A** | | | | | |
| CHAOS continuous | 0.02 (-0.05, 0.08) | 1.42 (-0.15, 2.98) | -0.03 (-0.08, 0.01) | 0.05 (-0.08, 0.17) | -0.02 (-0.15, 0.12) |
| **MODEL B** | | | | | |
| CHAOS Higher Vs. Lower | 0.11 (-0.21, 0.44) | 6.57 (-1.65, 14,79) | -0.20 (-0.43, 0.03) | 0.29 (-0.35, 0.93) | -0.17 (-0.85, 0.50) |
| **Girls (n = 173)** | | | | | |
| **MODEL A** | | | | | |
| CHAOS continuous | 0.01 (-0.08, 0.11) | 1.35 (-0.73, 3.44) | -0.10 (-0.15, -0.04)* | 0.10 (-0.07, 0.27) | 0.04 (-0.14, 0.23) |
| **MODEL B** | | | | | |
| CHAOS Higher Vs. Lower | 0.15 (-0.29, 0.59) | 6.11 (-5.21, 17.42) | -0.44 (-0.75, -0.14) | 0.74 (-0.13, 1.60) | 0.14 (-0.78, 1.07) |
| **Boys (n = 204)** | | | | | |
| **MODEL A** | | | | | |
| CHAOS continuous | 0.01 (-0.08, 0.12) | 1.56 (-0.82, 3.94) | 0.02 (-0.05, 0.08) | -0.01 (-0.20, 0.18) | -0.05 (-0.25, 0.14) |
| **MODEL B** | | | | | |
| CHAOS Higher Vs. Lower | 0.07 (-0.43, 0.57) | 9.27 (-3.22, 21.77) | -0.12 (-0.47, 0.23) | -0.19 (-1.17, 0.80) | -0.34 (-1.37, 0.70) |

Abbreviations: CHAOS, Confusion, Hubbub, and Order Scale; zBMI, body mass index z-scores based on the WHO reference norms; MVPA, moderate-to-vigorous physical activity.

Model A estimates associations for a 1-unit increase in the CHAOS score whereas Model B estimates associations for higher vs. lower CHAOS dichotomised at the 75th percentile. Models are adjusted for participants' age, parental education, household income, and family structure. Full sample models are additionally adjusted of sex. MVPA was transformed to normalise its distribution such that coefficients are interpreted as the % increase/decrease in MVPA for one unit increase in the independent variable.

* A statistically significant interaction was found by sex for the association between CHAOS (continues) and sleep duration, interaction term p-value = 0.019

specific models an association between CHAOS and sleep duration among girls was observed. For every additional 1-point increase in CHAOS, girls slept 0.10 hours less on average (Beta = -0.10; 95% CI -0.15, -0.04), with similar findings when examining CHAOS dichotomized. No associations were observed among boys or for other outcome variables. Lastly, in sensitivity analyses, findings were similar when dichotomising lifestyle behaviours as meeting vs not recommended levels and zBMI as normal weight vs overweight/obesity (results not shown).

## Discussion

This study is one of the few examining chaos in the family home environment in relation to multiple lifestyle behaviours and obesity in a sample of adolescents [8, 9]. Overall, we found little evidence for associations between chaos in the family environment, as reported by adolescents, and lifestyle behaviours or obesity, with the exception of a negative association with sleep duration among girls. On average, household chaos was low in this sample with girls reporting higher chaos compared to boys.

The association we found with sleep duration is consistent with those of two studies among Australian adolescents reporting associations between household chaos and sleep, namely shorter sleep duration and longer sleep onset latency [46, 47]. Likewise, a pilot study of African American adolescents (n = 26) found that adolescents were more likely to report the occurrence of sleep-disturbing activities and behaviours by other family members in increasingly chaotic households [48]. Other studies found chaos in the family home environment to be a mediator in the association between low socioeconomic status and sleep quality in adolescents

[18] and in children [19]. Associations between household chaos and sleep problems (e.g., bedtime resistance and sleep anxiety) have also been observed among preschool-aged children from low-income minority families [49]. In the current study, the association was only observed among girls and, contrary to some studies [18, 19, 48, 49], our sample included adolescents from mostly middle to high-income families.

On average, girls reported higher chaos levels compared to boys. Few studies have explored sex or gender differences in measured or perceived household chaos among adolescents. To the best of our knowledge, only one Pakistani study reported higher levels of household chaos among girls compared to boys in a sample of children aged 10 to 19 years, which is in agreement with our findings [50]. This may reflect gender differences whereby girls may rely more on social support from their immediate family and may be more involved in family chores and responsibilities [51, 52], thereby being more aware of sources of instability and disorganisation in the family environment. The QUALITY cohort did not include a measure of gender hence we report sex differences, but acknowledge that these differences likely reflect social rather than biological processes.

Overall low household chaos was reported by QUALITY cohort participants at the 3[rd] wave of data collection. To take part in the study, families needed to have the social and financial resources to participate in a full day research evaluation for the third wave of data collection, as well as the two prior data collection waves. Participants lost to follow up may be those with a higher level of chaos in the family home environment. In addition, the QUALITY cohort includes only White children with at least one parent with obesity. This may have limited the diversity of included participants in terms of sociodemographic profiles, and may have contributed to the absence of associations between household chaos, lifestyle behaviours and obesity in this study. Indeed, it is possible that household chaos influences these outcomes only after a given threshold of chaos has been reached.

In the systematic review by Marsh et al. on associations between household chaos and child health outcomes, authors report that household chaos has been found to mediate associations between low socioeconomic status and adverse child outcomes (e.g., low cortisol levels, socioemotional adjustment, academic achievement), although evidence for this mediation effect was not found for obesity outcomes [9]. None the less, studies focusing specifically on children from low socioeconomic status have observed associations between chaotic homes and excess weight status [25, 27]. Low income and socioeconomic status are well described determinants of child obesity in Canada [53]. Families with low income may experience higher levels of instability due to competing demands, frequent changes in housing and work or unreliable income, as well as higher levels of disorganisation due to crowding. One explanation for the absence of associations in the current study may be the overall affluence of included families with the majority being from middle- and high-income families.

Additional limitations related to the measurement of household chaos should be noted. Family home environment chaos was not measured at previous time points within the QUALITY cohort preventing us from examining longitudinal associations or cumulative effects of chaos over time. Although still widely used today, the CHAOS measurement tool was developed in 1994 and designed to be completed by parents [32]. The tool may have been somewhat confusing for contemporary adolescents. For example, the item "The telephone takes up a lot of our time at home" may be misunderstood as not including cellphones and tablets. A research assistant was present when the participants completed the questionnaire to help clarify questions for adolescents when needed. Moreover, asking adolescents to complete the questionnaire, as opposed to parents, may have yielded more honest responses, since there may be less imperative for socially desirable responses. One study found self-reported household chaos of parents and adolescents to be moderately correlated [54]. Still, it is unclear as to

whether the CHAOS questionnaire is robust enough to measure household chaos from an adolescent's perspective. Additionally, the CHAOS tool focuses mainly on the disorganization dimension of this construct and less so on the instability dimension [15, 17]. Some authors, based on the need to better capture the chaos construct and its subdomains, have argued in favor of using a more comprehensive household chaos measurement approach, notably to better capture the instability dimension [15], to include measurements of specific family routines [8], and to rely on systematic observations (e.g., direct observations and home tours) [15, 17]. Studies where specific family routines are assessed (e.g., sleep and meal-time routines and screen time limits) as indicators of household organisation, have reported a statistically significant associations with child overweight [8]. As opposed to chaotic households, organized homes have structured family routines and expectations, thus are thought to have positive influences on children and adolescents' lifestyle behaviours, weight status and overall development by promoting healthy, positive, and comforting interactions between family members [8]. Complementing the CHAOS tool with questionnaires that capture other elements of the household environment related to family organisation, routines and dynamics are needed to better understand the link with childhood obesity and related lifestyle behaviours.

## Conclusion

Although there is some research supporting a link between household chaos, lifestyle behaviours and obesity among children, few studies in adolescence exists. In this study, girls reported more household chaos than boys and only sleep duration was found to be shorter on average among girls who reported higher chaos compared to girls who reported lower chaos. This is one of the few studies on associations between household chaos, lifestyle behaviours and obesity among adolescents and adds to existing information on Quebec, Canada adolescents. Future studies should include more diverse samples of adolescents, examine sex and gender differences, and consider longitudinal data.

## Supporting information

**S1 Table. Item-score correlations for the 15 items included in the Confusion, Hubbub, and Order Scale (CHAOS), QUALITY cohort study (n = 377).**
(DOCX)

## Acknowledgments

Dr Marie Lambert (July 1952 –February 2012), pediatric geneticist and researcher, initiated the QUALITY cohort. Her leadership and devotion to QUALITY will always be remembered and appreciated. The cohort integrates members of TEAM PRODIGY, an inter-university research team including Université de Montréal, Concordia University, Centre INRS—Institut Armand-Frappier, Université Laval, and McGill University. The research team is grateful to all the children and their families who took part in this study, as well as the technicians, research assistants, and coordinators involved in the QUALITY cohort project.

## Author Contributions

**Conceptualization:** Andraea Van Hulst, Sujani Jayanetti, Tracie A. Barnett, Mélanie Henderson.

**Data curation:** Tracie A. Barnett, Mélanie Henderson.

**Formal analysis:** Andraea Van Hulst, Sujani Jayanetti, Ana Maria Sanson-Rosas, Lisa Kakinami.

**Investigation:** Andraea Van Hulst, Sujani Jayanetti, Ana Maria Sanson-Rosas, Marie-Josée Harbec, Lisa Kakinami.

**Methodology:** Andraea Van Hulst.

**Supervision:** Andraea Van Hulst.

**Writing – original draft:** Andraea Van Hulst, Sujani Jayanetti.

**Writing – review & editing:** Andraea Van Hulst, Sujani Jayanetti, Ana Maria Sanson-Rosas, Marie-Josée Harbec, Lisa Kakinami, Tracie A. Barnett, Mélanie Henderson.

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
