## [Decision Letter · Decision Letter 0]

4 Aug 2022

PONE-D-22-15790Adolescents’ reports of chaos within the family home environment: investigating associations with lifestyle behaviours and obesityPLOS ONE

Dear Dr. Hulst,

Thank you for submitting your manuscript to PLOS ONE. After careful consideration, we feel that it has merit but does not fully meet PLOS ONE’s publication criteria as it currently stands. Therefore, we invite you to submit a revised version of the manuscript that addresses the points raised during the review process.

We look forward to receiving your revised manuscript.

Kind regards,

Linglin Xie

Academic Editor

PLOS ONE

Journal Requirements:

"The QUALITY study (primary data collection) was funded by grants from CIHR (https://cihr-irsc.gc.ca/e/193.html) (#OHF-69442, #NMD-94067, #MOP-97853, #MOP-119512), HSFC (https://www.heartandstroke.ca) (#PG040291), and FRQS (https://frq.gouv.qc.ca). AVH is LK hold a Fonds de la recherche en santé du Québec (FRQS) Junior 1 award and MH holds a Junior 2 award from the same organization. The secondary analysis presented herein did not receive any funding."

Reviewers' comments:

Reviewer's Responses to Questions

**Comments to the Author**

1. Is the manuscript technically sound, and do the data support the conclusions?

Reviewer #1: Yes

2. Has the statistical analysis been performed appropriately and rigorously? 

Reviewer #1: Yes

3. Have the authors made all data underlying the findings in their manuscript fully available?

Reviewer #1: No

4. Is the manuscript presented in an intelligible fashion and written in standard English?

Reviewer #1: Yes

5. Review Comments to the Author

Reviewer #1: In the manuscript, the authors performed comprehensive analyses between household chaos and lifestyle behaviors or obesity in adolescents. Only a negative association with sleep duration among adolescent girls was observed, mainly due to a few intrinsic limitations given the samples being collected. Please refer to the comments below.

Major Comment

The authors mentioned that household chaos might have a strong influence on the eating habits of young children. Therefore, diet or unhealthy diet due to household chaos might be directly associated with childhood overweight/obesity. Is it not fully considered in the survey? Since there is no strong association between vegetable or fruits intake and CHAOS, I wonder if the definition of diet category is too vague. Measurements like calories (intake) could be potentially important attributes.

One problem I have with the study is that the samples have a relatively low to moderate household chaos score on average and this might be the reason for the lack of evidence for associations between chaos and obesity (One of the major limitations of study as the authors mentioned). Would the authors consider adding more samples with higher chaos if available?

In addition to those limitations of the samples in this study, I do want to question the robustness of the (or design of) the CHAOS questionnaire. At least based on the outcome, the results do not substantially coalign with the significance of study.

Additionally, since the results might be sensitive to income level, have the authors considered collecting data or samples from low-income families? It would be interesting to analyze the proposed threshold of chaos for more evident influences.

Minor Comment

Please reformat the references and follow the guidelines properly. I noticed some issues, for instance, no URL should be included.

6. PLOS authors have the option to publish the peer review history of their article (what does this mean?). If published, this will include your full peer review and any attached files.

Reviewer #1: No

---

## [Author Response · Author response to Decision Letter 0]

10 Nov 2022

Rebuttal Letter to PLOS ONE 

Manuscript Number: PONE-D-22-15790

Title: Adolescents’ reports of chaos within the family home environment: investigating associations with lifestyle behaviours and obesity

Dear Dr. Linglin Xie, 

We wish to thank the editorial team and reviewers for their helpful comments to the above-mentioned manuscript and are pleased to provide you with a revised manuscript. 

A detailed response to each comment is provided below. We hope that these revisions will be found suitable for our manuscript to be considered for publication in PLOS ONE.

Editorial comments

Comment 1. Please ensure that your manuscript meets PLOS ONE's style requirements, including those for file naming. The PLOS ONE style templates can be found at 

Response: We have reviewed the journal’s style requirements and have made all required changes as per the instructions provided. 

Comment 2. Thank you for stating the following financial disclosure: 

"The QUALITY study (primary data collection) was funded by grants from CIHR (https://cihr-irsc.gc.ca/e/193.html) (#OHF-69442, #NMD-94067, #MOP-97853, #MOP-119512), HSFC (https://www.heartandstroke.ca) (#PG040291), and FRQS (https://frq.gouv.qc.ca). AVH is LK hold a Fonds de la recherche en santé du Québec (FRQS) Junior 1 award and MH holds a Junior 2 award from the same organization. The secondary analysis presented herein did not receive any funding."

Response: We have added information on the role of the funders as: “The funders had no role in study design, data collection and analysis, decision to publish, or preparation of the manuscript.”

Comment 3. In your Data Availability statement, you have not specified where the minimal data set underlying the results described in your manuscript can be found. PLOS defines a study's minimal data set as the underlying data used to reach the conclusions drawn in the manuscript and any additional data required to replicate the reported study findings in their entirety. All PLOS journals require that the minimal data set be made fully available. For more information about our data policy, please see http://journals.plos.org/plosone/s/data-availability.

Response: For ethical reasons, data from study participants cannot be shared openly as they include potential identifying participant information. We commit to making individual participant data from the QUALITY Cohort study that were used in the current analysis available as per the following conditions:

Will individual participant data be available (including data dictionaries)? Yes

What data in particular will be shared? Individual participant data that underlie the results reported in this article, after de-identification (text, tables, figures, and appendices)

What other documents will be available? Study protocol, analytic code

When will data be available (start and end dates)? Beginning 3 months and ending 6 months following article publication

With whom? Investigators whose proposed use of the data has been approved by the executive committee of the QUALITY cohort 

For what types of analyses? To replicate findings from the current project

By what mechanism will data be made available? Requests should be directed to melanie.henderson.hsj@gmail.com (QUALITY Cohort Study PI); to gain access, data requestors will need to sign a data access agreement

Comment 4. Please include captions for your Supporting Information files at the end of your manuscript, and update any in-text citations to match accordingly. Please see our Supporting Information guidelines for more information: http://journals.plos.org/plosone/s/supporting-information.

Response: We have made the requested changes.

Reviewers' comments:

Comment 5. Reviewer #1: In the manuscript, the authors performed comprehensive analyses between household chaos and lifestyle behaviors or obesity in adolescents. Only a negative association with sleep duration among adolescent girls was observed, mainly due to a few intrinsic limitations given the samples being collected. Please refer to the comments below.

Major Comment: The authors mentioned that household chaos might have a strong influence on the eating habits of young children. Therefore, diet or unhealthy diet due to household chaos might be directly associated with childhood overweight/obesity. Is it not fully considered in the survey? Since there is no strong association between vegetable or fruits intake and CHAOS, I wonder if the definition of diet category is too vague. Measurements like calories (intake) could be potentially important attributes.

Response: We thank the reviewer for this comment. Indeed, given the absence of any associations between household chaos with dietary intake as measured by habitual servings of vegetables and fruit, and with obesity as measured by BMI z-score, our data do not support the potential mediating role of vegetable and fruit intake in the association between household chaos and adolescent BMI. As such this was not further investigated. 

We agree with the reviewer that accurately assessing diet quality is challenging. For this study, we used mean daily servings of vegetables and fruits obtained from three 24-hour diet recalls as an indicator of overall diet quality. Although calorie intake may be an alternative measure, we believe using it as an indicator of diet quality may be problematic. 

First, measurement error is likely in total calorie intake. In previous work using QUALITY cohort data, we have shown that misreporting of energy intake is common among participants. In particular, BMI z-score was an important predictor of energy intake underreporting (Suissa K, Benedetti A, Henderson M, Gray-Donald K, Paradis G. The Cardiometabolic Risk Profile of Underreporters of Energy Intake Differs from That of Adequate Reporters among Children at Risk of Obesity. J Nutr. 2019 Jan 1;149(1):123-130). 

Second, total calorie intake is dependent on other variables such as age, sex, physical activity level and weight status. In contrast, recommendations regarding servings of vegetable and fruit intake for adolescents are standard, regardless of these other variables. 

Third, we opted for vegetable and fruit intake as it has previously been used in Quebec (Canada) as a measure of diet quality (for example: https://statistique.quebec.ca/fr/fichier/enquete-quebecoise-sur-la-sante-des-jeunes-du-secondaire-2016-2017-resultats-de-la-deuxieme-edition-tome-3-la-sante-physique-et-les-habitudes-de-vie-des-jeunes.pdf). Moreover, vegetable and fruit intake has been linked prospectively to more optimal cardiometabolic profiles including in the QUALITY cohort (Van Hulst A, Paradis G, Harnois-Leblanc S, Benedetti A, Drapeau V, Henderson M. Lowering Saturated Fat and Increasing Vegetable and Fruit Intake May Increase Insulin Sensitivity 2 Years Later in Children with a Family History of Obesity. J Nutr. 2018 Nov 1;148(11):1838-1844.)

To clarify this, we have added the following sentence in the methods section (clean version lines 196-197) and added a reference to justify our use of vegetable and fruit intake as a measure of diet quality. 

“Daily average servings of vegetables and fruits intake was considered in this study as an indicator of overall diet quality [43].”

Comment 6: One problem I have with the study is that the samples have a relatively low to moderate household chaos score on average and this might be the reason for the lack of evidence for associations between chaos and obesity (One of the major limitations of study as the authors mentioned). Would the authors consider adding more samples with higher chaos if available?

Response: Given that this study relies on a secondary data analysis of the 3rd wave of already collected data from the QUALITY cohort, it was not possible to recruit additional participants to this study. However, as mentioned by the reviewer, we acknowledge this limitation extensively in our discussion (clean version lines 287-296). 

Comment 7: In addition to those limitations of the samples in this study, I do want to question the robustness of the (or design of) the CHAOS questionnaire. At least based on the outcome, the results do not substantially coalign with the significance of study.

Response: We agree with this reviewer comment. Although being one of the most widely used tools in published studies on household chaos, its ability to adequately capture we question the validity of the tool’s ability to capture instability, turbulence and disorganisation within the household has been questioned. We refer to this literature in the discussion, notably with regards to the need for measurement tools that better capture the instability dimension of household chaos (clean version lines 309-336).

Comment 8: Additionally, since the results might be sensitive to income level, have the authors considered collecting data or samples from low-income families? It would be interesting to analyze the proposed threshold of chaos for more evident influences.

Response: As per our response to comment 6, we relied solely on existing data from a Quebec-based cohort study to examine associations between household chaos and lifestyle behaviours/obesity in adolescents. As such, we were not able to add data from low-income families to this specific study. We mention this in the discussion and refer to extant literature that have focused on the household chaos in samples of children and adolescents from lower socio-economic backgrounds (clean version lines 301-308).

Comment 9: Minor Comment: Please reformat the references and follow the guidelines properly. I noticed some issues, for instance, no URL should be included.

Response: We have made the corrections to the references.

We hope that the modifications brought to the manuscript will meet the editorial team and reviewers’ satisfaction and qualify for publication.

Sincerely, 

Andraea Van Hulst, PhD, RN

Assistant Professor

McGill University Ingram School of Nursing

680 Sherbrooke West, Office 1833

Montreal, QC, Canada H3A 2M7

---

## [Decision Letter · Decision Letter 1]

9 Jan 2023

Adolescents’ reports of chaos within the family home environment: investigating associations with lifestyle behaviours and obesity

PONE-D-22-15790R1

Dear Dr. Hulst,

We’re pleased to inform you that your manuscript has been judged scientifically suitable for publication and will be formally accepted for publication once it meets all outstanding technical requirements.

Kind regards,

Linglin Xie

Academic Editor

PLOS ONE

Additional Editor Comments (optional):

Reviewers' comments:

Reviewer's Responses to Questions

**Comments to the Author**

1. If the authors have adequately addressed your comments raised in a previous round of review and you feel that this manuscript is now acceptable for publication, you may indicate that here to bypass the “Comments to the Author” section, enter your conflict of interest statement in the “Confidential to Editor” section, and submit your "Accept" recommendation.

Reviewer #1: All comments have been addressed

2. Is the manuscript technically sound, and do the data support the conclusions?

Reviewer #1: Yes

3. Has the statistical analysis been performed appropriately and rigorously? 

Reviewer #1: Yes

4. Have the authors made all data underlying the findings in their manuscript fully available?

Reviewer #1: Yes

5. Is the manuscript presented in an intelligible fashion and written in standard English?

Reviewer #1: Yes

6. Review Comments to the Author

Reviewer #1: The authors have addressed my previous comments with adequate details and explanations.

(Minor comment) In the clean version of the revised manuscript, line number seems missing after discussion section.

For the new reference 43, please add doi if applicable: doi: 10.1080/10408398.2019.1632258.

7. PLOS authors have the option to publish the peer review history of their article (what does this mean?). If published, this will include your full peer review and any attached files.

Reviewer #1: No

---

## [Editor Report · Acceptance letter]

16 Jan 2023

PONE-D-22-15790R1 

Adolescents’ reports of chaos within the family home environment: investigating associations with lifestyle behaviours and obesity 

Dear Dr. Van Hulst:

I'm pleased to inform you that your manuscript has been deemed suitable for publication in PLOS ONE. Congratulations! Your manuscript is now with our production department. 

Kind regards, 

on behalf of

Dr. Linglin Xie 

Academic Editor

PLOS ONE